# Independent Predictors of Circulating Trimethylamine N-Oxide (TMAO) and Resistin Levels in Subjects with Obesity: Associations with Carotid Intima-Media Thickness and Metabolic Parameters

**DOI:** 10.3390/nu17050798

**Published:** 2025-02-26

**Authors:** Denisa Pescari, Monica Simina Mihuta, Andreea Bena, Dana Stoian

**Affiliations:** 1Department of Doctoral Studies, “Victor Babeș” University of Medicine and Pharmacy, 300041 Timisoara, Romania; denisa.bostina@umft.ro; 2Center for Molecular Research in Nephrology and Vascular Disease, “Victor Babeș” University of Medicine and Pharmacy, 300041 Timisoara, Romania; stoian.dana@umft.ro; 3Discipline of Endocrinology, Second Department of Internal Medicine, “Victor Babeș” University of Medicine and Pharmacy, 300041 Timisoara, Romania; borlea.andreea@umft.ro

**Keywords:** TMAO, resistin, carotid intima-media thickness, obesity, cardiovascular risk

## Abstract

**Background:** Obesity contributes to cardiometabolic risk, including subclinical atherosclerosis and insulin resistance. This study examines the predictive roles of trimethylamine N-oxide (TMAO) and resistin in relation to carotid intima-media thickness and metabolic parameters; **Methods**: Sixty adults (18–71 years) with varying body weights were assessed for body composition, subclinical atherosclerosis, and blood biomarkers, including TMAO and resistin; **Results**: TMAO correlated strongly with CIMT (r = 0.674, *p* < 0.001), indicating its role in subclinical atherosclerosis. Logistic regression identified TMAO (threshold 380; AUC = 0.880, accuracy = 91.7%) as a predictor of cardiometabolic risk. Resistin was associated with CIMT, WHR, and total cholesterol, inversely linked to LDL cholesterol (*p* = 0.003). Less active participants exhibited higher TMAO (*p* = 0.001) and resistin (*p* = 0.02). Family histories of obesity and diabetes correlated with elevated TMAO, while resistin linked to shorter sleep duration and diabetes history, highlighting their importance in obesity-related cardiometabolic risks; **Conclusions**: TMAO is strongly linked to abdominal fat, insulin resistance, and subclinical atherosclerosis, while resistin is associated with lipid metabolism and aging. Their combined assessment enhances the prediction of obesity-related cardiometabolic risk, supporting their role in risk stratification and targeted interventions.

## 1. Introduction

The increasing prevalence of metabolic disorders, primarily driven by overweight, obesity, and diabetes, represents a major global public health challenge, impacting both industrialized and developing nations [1]. Obesity is a multifactorial condition influenced by genetic, epigenetic, and environmental factors, including dietary patterns, physical activity levels, and exposure to pollutants [2]. According to the World Health Organization, as of 2016, 13% of adults were classified as obese, while 39% were overweight [3,4]. Furthermore, the global prevalence of overweight individuals has doubled since 1990, underscoring the urgent need for effective prevention and intervention strategies.

Obesity is associated with a broad range of adverse health outcomes, including an increased risk of cardiovascular diseases and insulin resistance [5]. Beyond these established links, obesity significantly contributes to global morbidity and mortality, predisposing individuals to type 2 diabetes, certain cancers, and respiratory disorders, ultimately reducing life expectancy and quality of life [5,6,7,8]. While BMI remains the most widely used tool for assessing nutritional status at the population level [9,10], it primarily reflects overall adiposity without providing detailed body composition data. Importantly, adipose tissue distribution varies with age and sex, independent of BMI values [11,12]. Recent findings indicate that BMI assessment in individuals with overweight and obesity exhibits greater predictive accuracy for metabolic conditions than models based solely on adiposity measurement [13].

Anthropometric parameters such as waist circumference (WC) and waist-to-hip ratio (WHR) are widely recognized as key indicators of cardiometabolic risk [14]. WC, a direct measure of central obesity, is strongly associated with insulin resistance, dyslipidemia, and systemic inflammation, all of which contribute to the pathogenesis of cardiometabolic diseases [15]. Notably, increased WC is a significant predictor of cardiometabolic outcomes, even among individuals with normal BMI, underscoring its superiority in risk assessment [16]. WHR, which reflects fat distribution, is strongly linked to increased cardiometabolic risk and all-cause mortality [17]. Central adiposity, as indicated by a high WHR, correlates with insulin resistance, impaired glucose metabolism, and dysregulated lipid profiles, significantly elevating the risk of type 2 diabetes and cardiovascular events [18]. WHR has been shown to independently predict cardiovascular risk, even after adjusting for traditional risk factors such as BMI, age, and smoking status, reinforcing its clinical utility in identifying high-risk individuals [19]. These parameters provide a practical and cost-effective means for clinicians to assess and monitor cardiometabolic health, facilitating early intervention in at-risk populations.

Atherosclerosis-driven arterial dysfunction leads to atheromatous plaque formation and cardiovascular events in both adults and children [20]. Given the increasing global burden of cardiovascular disease, early detection of subclinical atherosclerosis in obesity is crucial for timely intervention. Lipid accumulation, inflammation, and immune cell infiltration are primary drivers of atherosclerosis, with LDL-c playing a central role in endothelial lipid deposition [21,22,23,24]. Obesity exacerbates atherosclerosis by impairing nitric oxide (NO) production, disrupting vascular homeostasis, and increasing insulin resistance, all of which contribute to endothelial dysfunction [2]. Central obesity, particularly visceral fat, is a significant risk factor for atheromatous plaque development [25].

Carotid intima-media thickness (CIMT) is a widely validated marker of subclinical atherosclerosis, offering a non-invasive assessment of early arterial changes and cardiovascular risk, even in asymptomatic individuals [26,27]. Increased CIMT in obesity is driven by chronic low-grade inflammation, insulin resistance, and endothelial dysfunction [28]. Visceral adipose tissue secretes pro-inflammatory cytokines, including TNF-α, IL-6, and leptin, promoting vascular inflammation and oxidative stress [29]. These mediators facilitate leukocyte recruitment, adhesion molecule expression, and arterial wall thickening [30,31]. Obesity-associated dyslipidemia, characterized by elevated triglycerides and small, dense LDL-c particles, further promotes atherosclerosis and CIMT elevation [32]. Obesity-related CIMT elevation in adolescents suggests early vascular aging and increased lifelong cardiovascular risk [33]. Central obesity correlates more strongly with CIMT than subcutaneous fat, emphasizing the impact of fat distribution [34]. Normal CIMT in adults ranges from 0.5 to 0.9 mm, with values above 1.0 mm indicating subclinical atherosclerosis [35]. Even without overt cardiovascular disease, obesity is associated with increased CIMT, reflecting early vascular damage [36]. CIMT is a valuable tool for cardiovascular risk assessment, particularly in obese individuals with hypertension or diabetes, supporting early detection and targeted interventions. Longitudinal CIMT monitoring further informs the effectiveness of therapies aimed at reversing atherosclerotic changes in obesity [37].

TMAO is a metabolite produced in the liver from trimethylamine (TMA), which is generated by gut microbiota through the metabolism of dietary nutrients such as choline, phosphatidylcholine, and L-carnitine [38]. Recent research has highlighted the pivotal role of TMAO in the pathophysiology of cardiometabolic diseases, particularly in promoting cardiovascular disease and contributing to the risk of cardiometabolic disorders. Elevated plasma TMAO levels have been shown to accelerate atherosclerosis by enhancing the deposition of cholesterol in the arterial wall and promoting pro-inflammatory signaling in vascular endothelial cells, which results in increased plaque formation and plaque instability [39]. Additionally, TMAO has been found to impair reverse cholesterol transport, a process crucial for reducing excess cholesterol from peripheral tissues to the liver for excretion, further exacerbating the development of atherosclerotic lesions [40]. Beyond its direct impact on cardiovascular health, TMAO has been implicated in the progression of insulin resistance, which is a fundamental component of metabolic syndrome and type 2 diabetes. Importantly, elevated TMAO levels have been strongly correlated with an increased incidence of major adverse cardiovascular events, such as myocardial infarction, stroke, and heart failure, even when accounting for traditional cardiovascular risk factors [41]. The identification of TMAO as an independent predictor of cardiovascular risk underscores its potential utility as a biomarker for early detection and risk stratification in individuals with or at risk of cardiometabolic diseases [42].

Resistin, a pro-inflammatory adipokine secreted by immune cells in humans, plays a key role in cardiometabolic diseases such as type 2 diabetes, atherosclerosis, and hypertension [43]. Elevated resistin levels are strongly linked to insulin resistance, as they impair insulin signaling, increase hepatic glucose production, and reduce peripheral glucose uptake, contributing to hyperglycemia and metabolic syndrome [44]. Elevated plasma resistin levels correlate with major adverse cardiovascular events, including coronary artery disease and heart failure, suggesting its potential as a biomarker for cardiovascular risk. Additionally, resistin contributes to hypertension by promoting vasoconstriction and arterial stiffness [45].

The aim of this cross-sectional observational study is to comprehensively analyze the independent determinants of circulating TMAO and resistin levels in individuals with obesity and to investigate their associations with CIMT and a broad spectrum of metabolic parameters, including BMI, bioimpedance parameters, lipid profile abnormalities, glycemic control—fasting blood glucose (FBG), HbA1c, and insulin resistance by homeostasis model assessment-estimated insulin resistance (HOMA-IR). This study seeks to elucidate the mechanistic pathways linking TMAO and resistin with obesity-induced vascular alterations and metabolic dysregulation, specifically focusing on their potential contributions to subclinical atherosclerosis, as indicated by CIMT, and the overall cardiometabolic risk profile in this population.

## 2. Materials and Methods

The cross-sectional observational study was conducted in the endocrinology unit from October 2023 to June 2024. A total of 60 adult participants were enrolled, comprising 13 males (21.62%) and 47 females (78.48%), with ages ranging from 18 to 71 years and a mean age of 37.30 ± 14.82 years. The participants sought to evaluate their nutritional status and initiate a personalized nutrition plan aimed at long-term lifestyle optimization. All participants provided informed consent. This study adhered to the ethical standards of the Helsinki Declaration and was approved by the Scientific Research Ethics Committee (CECS) of the “Victor Babeș” University of Medicine and Pharmacy Timișoara (No. 69/03.10.2022).

Following the assessment of nutritional status, the cohort was divided into study groups based on the severity of excess body weight, specifically the normal weight group or control group (BMI between 18.5 and 24.9 kg/m^2^), the overweight group (BMI between 25 and 29.9 kg/m^2^), and the obesity group (BMI over 30 kg/m^2^). The latter group was further classified according to the severity of obesity: grade I (BMI between 30 and 34.9 kg/m^2^), grade II (BMI between 35 and 39.9 kg/m^2^), and grade III (BMI over 40 kg/m^2^). Additionally, the control group included participants with normal weight who had no additional cardiometabolic risk factors, were non-smokers, had no personal or family history of cardiometabolic disease, and did not consume alcohol.

Additionally, subgroups were created based on the reference values for TMAO, specifically a threshold of 380 µg/L. The threshold of 380 µg/L (4.9 µM) for TMAO was selected based on laboratory validation and aligns with previously reported cutoff values associated with increased cardiovascular risk, as demonstrated in multiple studies linking TMAO levels above 2–4 µM with heightened cardiovascular morbidity and mortality risk [40,46,47,48,49]. Consequently, subjects were divided into a high cardiometabolic risk group, characterized by TMAO levels exceeding this threshold, and a group with no added risk, defined by optimal TMAO levels below 380 µg/L.

### 2.1. Patient Inclusion and Exclusion Criteria

Inclusion criteria:-Adults over 18 years old, both male and female.-Individuals with excess body weight seeking a hypocaloric dietary program.-Participants with a first-degree family history of metabolic or cardiovascular diseases.-Both smokers and non-smokers.-Participants with a consistent diet including fish, meat, grains, and eggs in the past six months.-Control group: individuals with optimal weight and no personal or family history of cardiometabolic diseases.-Subjects who completed anamnesis, clinical, and nutritional evaluation and provided written informed consent.

Exclusion criteria: To ensure that observed differences in TMAO and resistin levels were not confounded by pre-existing metabolic dysfunction, we selected a control group with no cardiometabolic risk factors. This approach aimed to establish a clear reference point for comparison with overweight and obese individuals.-Children and adolescents.-Individuals following vegetarian, vegan, or gluten-free diets.-Use of dietary supplements containing choline.-Diagnosed secondary causes of obesity.-History of nutritional interventions for weight loss in the past 12 months.-Use of anti-obesity medications in the last 16 weeks.-History of metabolic surgery in the past five years.-Use of medications affecting TMAO and resistin levels, including the following:-Lipid-lowering agents (statins, fibrates, and ezetimibe) [39,50].-Antibiotics and probiotics (within two months before TMAO measurement). [51].-Antidiabetic drugs (metformin and thiazolidinediones) [52].-Nonsteroidal anti-inflammatory drugs and corticosteroids.-Hormone replacement therapy and oral contraceptives [53].-Presence of psychiatric disorders.-Impaired kidney function or chronic kidney disease (eGFR < 90 mL/min/1.73 m^2^, CKD-EPI).

### 2.2. Patient Complete Evaluation

Before any procedures included in this study were conducted, each patient was fully informed regarding the details of the research, including the clinical and paraclinical examinations. No procedures were implemented until the informed consent form was thoroughly explained, accepted, and signed. A comprehensive anamnesis was performed during the initial consultation, covering demographic data, family medical history (specifically first-degree relatives with excess body weight), personal cardiometabolic history, dietary habits, and behavioral factors. Additionally, each participant underwent a full clinical examination at the initial visit. The non-invasive methods used in our investigation included bilateral ultrasonographic measurement of carotid intima-media thickness and bioelectrical impedance analysis (BIA) to estimate segmental body composition. The initial evaluation of this study included the following parameters:

#### 2.2.1. Sociodemographic and Behavioral Characteristics

As a result of conducting a comprehensive anamnesis, the following variables related to the participants’ daily routine were included in this section:

Smoking status: classified as positive if the individual has smoked at least one cigarette per day for a minimum duration of one year.Sleep pattern: Sleep duration was evaluated for each participant, with a nightly duration of less than 7 h categorized as sleep deprivation or an inadequate sleep schedule [54,55].Activity engagement level: To be classified outside the sedentary category, participants were required to engage in sustained physical activity for a minimum of 30 min per day or 150 min per week, exceeding the combined threshold of active and basal activity levels [56].Alcohol intake: Participants self-reported their alcohol intake, measured in units, where one unit was equivalent to 10 mL of pure ethanol. The following definitions were used: two units corresponded to a pint or can of beer, one unit to a 25 mL shot of spirits, and one unit to a standard 175 mL glass of wine. Participants consuming more than two units of alcohol per day were categorized as “excessive drinkers”, while those who had never consumed alcohol were classified as “abstainers” [57].

#### 2.2.2. Nutritional Assessment

The nutritional status of each participant was evaluated using BMI, a commonly employed and cost-efficient metric. BMI was calculated based on the formula: BMI = weight (kg)/height^2^ (m^2^) [26,58]. Additionally, other anthropometric parameters such as WC and WHR were measured for each participant.

Body weight assessment: Body weight was measured using a mechanically certified scale with metrological validation, capable of recording up to 200 kg. Participants were instructed to maintain an upright posture on the scale while lightly clothed.Height assessment: Height was measured using a calibrated, wall-mounted stadiometer. Participants were instructed to stand in an upright position on the platform, barefoot, to ensure precise measurement.

### 2.3. Personal and Family Medical History

#### 2.3.1. Family Medical History and Conditions

A comprehensive assessment of family medical history, specifically targeting cardiometabolic disorders, was conducted for each participant. Key conditions of interest, particularly in first-degree relatives, included obesity, type 2 diabetes, and cardiovascular diseases, including essential hypertension, acute myocardial infarction, and stroke.

#### 2.3.2. Medical History and Health Conditions

A structured evaluation of each participant’s personal medical history was conducted through targeted questioning, supported by medical records provided during the initial consultation for verification. This process facilitated the exclusion of specific conditions. The primary focus was on cardiovascular and metabolic disorders, including diagnosed essential hypertension or antihypertensive treatment, prediabetes, type 2 diabetes, metabolic syndrome, lipid profile modifications, asymptomatic hyperuricemia, and vitamin D status.

### 2.4. Laboratory Assay

Blood samples were obtained within one week following the physical examination, with collection times standardized between 7:30 and 8:30 a.m. Participants adhered to a fasting period of at least 12 h prior to sample collection. All serum parameter analyses were conducted in an accredited laboratory to ensure accuracy and compliance with standard protocols. Serum TMAO concentrations were measured using liquid chromatography-mass spectrometry (LC-MS). The reference values for serum TMAO were defined as follows: normal (<270 μg/L), borderline (≥270 μg/L to <380 μg/L), and elevated (≥380 μg/L) [59]. At this stage, serum resistin levels were also measured, with reference values stratified by gender as follows: for women, optimal values ranged between 3.7 and 13.6 ng/mL, and for men, the optimal range was 3.3–11.7 ng/mL [60]. Additional serum parameters measured included the lipid panel: TC (mg/dL), LDL-c (mg/dL), high-density lipoprotein cholesterol (HDL-c) (mg/dL), TG (mg/dL), uric acid (mg/dL), glycated hemoglobin (HbA1c) (%), HOMA-IR, FBG (mg/dL), and vitamin D levels. Additionally, TSH, FT4, and liver enzymes, namely, AST and ALT, were measured.

### 2.5. Bioimpedance Measurement Parameters

All participants in this study underwent an initial assessment of their nutritional status through bioimpedance analysis using the Tanita Body Composition Analyzer BC-418 MA III (T5896, Tokyo, Japan). This analysis specifically targeted the quantification and distribution of adipose tissue. A comprehensive evaluation of total body composition was conducted using a constant high-frequency electrical current (50 kHz, 500 μA) and employing a tetra-polar, eight-point tactile electrode system. Participants were instructed to stand in an upright position and maintain contact with the analyzer’s handles, ensuring appropriate connection with the eight electrodes, two for each foot and hand [61]. During the bioelectrical impedance analysis, a low-level electrical current was applied to the body, and impedance—characterized as the opposition to the flow of this current—was measured. This impedance reflects the body’s resistance and reactance, providing insight into various components of body composition, such as fat mass, lean tissue, and total body water [61,62]. The entire procedure was completed in approximately three minutes, with the results systematically documented and subsequently explained to each participant. Previous research has shown that, in clinical environments, the Tanita Body Fat Monitor exhibits an accuracy margin within ±5% when compared to the reference standard for body composition assessment, dual-energy x-ray absorptiometry (DXA) [62,63,64]. This degree of accuracy supports the device’s reliability as a practical tool for body composition evaluation in both research and clinical settings [65]. Additionally, Tanita reports that its method is the most accessible and convenient for predicting body composition with accuracy. The parameters measured by this technique were classified into the following categories: percentage of adipose tissue (%), percentage of trunk adipose mass (%), fat-free mass (kg), percentage of muscle tissue (%), basal metabolic rate (BMR), and percentage of total body water (%).

### 2.6. Carotid Intima-Media Thickness Assessment

The Aixplorer MACH 30 ultrasound system (SuperSonic Imagine, Aix-en-Provence, France) was used to conduct an ultrasonographic assessment of carotid intima-media thickness. A certified and highly experienced sonographer conducted carotid ultrasonography on each participant included in this study. The procedure involved carefully positioning each subject to ensure optimal visualization of the carotid artery and applying a conductive gel to facilitate sound wave transmission. The sonographer then meticulously scanned the carotid arteries, using appropriate transducer settings based on individual anatomical characteristics, such as neck structure and adipose tissue distribution: SL 18-5 (5–18 MHz) or SL 10-2 (2–10 MHz). The CIMT values were automatically computed by the advanced software embedded within the ultrasound system (SuperSonic Imagine 3.0, Aix-en-Provence, France). This software analyzes the ultrasound images in real-time, providing precise measurements of the intima-media thickness without requiring manual calculation, ensuring both accuracy and efficiency in the evaluation process. The ultrasound images were captured during the end-diastolic phase, identified by the occurrence of the R wave on the electrocardiogram, ensuring consistency in arterial relaxation and minimizing variability in measurements [66]. For each participant, six separate CIMT measurements were taken, with three measurements performed on both the left and right carotid arteries. The measurements were carefully averaged to produce a mean CIMT value, which was subsequently used for analysis in this study. This approach helped to enhance the precision and reliability of the data by accounting for natural variations in arterial thickness across different locations and ensuring that the final CIMT value represented an accurate reflection of each patient’s vascular status. To obtain optimal visualization of the right and left common carotid arteries, the subject is positioned in a supine position and instructed to extend their neck backward as far as comfortably possible, enhancing exposure of the cervical region. Additionally, the examiner carefully chooses the most suitable ultrasound transducer, ensuring that the correct frequency range is used to maximize image clarity and resolution for accurate assessment of the CIMT. The scanning procedure begins with a transverse approach, starting at the clavicle and moving upward along the neck to identify the carotid bulb and the bifurcation of the common carotid artery into the internal and external carotid arteries. Once the carotid bulb is located, the examiner transitions to longitudinal scanning, allowing for a detailed assessment of the arterial walls and a more precise measurement of the CIMT along the length of the artery. This systematic approach ensures thorough visualization of both the structure and flow characteristics within the carotid arteries. In this section, the carotid bulb is visualized on the left side of the ultrasound screen. Measurements are taken from the posterior wall of the carotid artery, specifically 1 to 2 cm distal to the carotid bulb. This location is chosen to avoid the geometrical irregularities of the bulb itself and to obtain accurate and consistent CIMT values from a more uniform segment of the arterial wall [67]. At end-diastole, the image is frozen, and the software automatically measures the CIMT in the examiner’s selected region of interest, ensuring consistency and precision [68,69].

Figure 1 illustrates an example of CIMT evaluation for the left carotid artery obtained using this method in a normoweight subject, whose CIMT value falls within optimal limits. In contrast, Figure 2 depicts an elevated left CIMT value of 1 mm, observed in a patient with grade II obesity. It is important to note that personal data of the evaluated subjects are not displayed to ensure confidentiality.

### 2.7. Statistical Analysis

In this study, we employed a variety of statistical tools to analyze the data and draw meaningful conclusions. Given that the normality of numerical variables was assessed using the Shapiro–Wilk test, which revealed that all variables followed a non-Gaussian distribution, we applied non-parametric methods throughout the analysis. Numerical variables were summarized as medians with interquartile ranges (IQR), while categorical variables were presented as proportions. To compare differences between two groups, we employed the Mann–Whitney U test, while for comparisons across multiple groups, the Kruskal–Wallis test was used. Post-hoc pairwise comparisons for Kruskal–Wallis were adjusted using the Dunn test with *p*-value adjustments using the Holm method (pHolm adjustment) to control for multiple comparisons. Correlation analyses were conducted using Spearman’s correlation for TMAO, given its ordinal nature, and Kendall’s tau for resistin, due to the presence of many tied ranks, ensuring a more accurate rank-based correlation analysis. Effect size was evaluated using Cohen’s criteria, which classifies the strength of the association as small (around 0.1), medium (around 0.3), or large (around 0.5). To identify significant independent predictors of TMAO and resistin levels, we employed multiple linear regression analysis. Variable selection for the models was performed using the backward elimination method, systematically removing non-significant predictors to achieve an optimal model. The best model was selected based on the Akaike Information Criterion (AIC) and Bayesian Information Criterion (BIC) to ensure the most parsimonious fit. Model performance was evaluated using adjusted R-squared to account for the proportion of variance explained by the model. For assessing cardiometabolic risk, multiple logistic regression was applied. The selection of variables followed backward elimination, and the model’s performance was evaluated using Nagelkerke’s R-squared (R^2^ Nagelkerke) to assess explained variance. Additionally, AUROC (Area Under the Receiver Operating Characteristic Curve) was used to quantify the model’s discriminatory ability. To examine the interaction between TMAO and resistin in predicting insulin resistance and obesity, we conducted logistic regression models with interaction terms, followed by interaction plot analyses to visualize the modifying effect of one biomarker on the relationship between the other and the outcome variables.

The results of all statistical analyses are presented in tables and graphical visualizations. A *p*-value < 0.05 was considered statistically significant, with a 95% confidence interval applied to all tests. All statistical analyses and data processing were conducted using R (version 4.3.0) software (R Core Team, 2024).

## 3. Results

### 3.1. Impact of BMI on TMAO, Resistin, and Associated Physiological and Metabolic Parameters

Significant trends were observed in TMAO and resistin levels, which increased with BMI. Median TMAO levels rose from 122.00 in the Normal weight group to 254.00 in the Obesity III group (*p* < 0.001), while resistin levels increased from 5.50 to 9.00 (*p* = 0.001). CIMT also increased with BMI, with right CIMT values rising from 0.60 in the control group to 0.88 in the Obesity III group (*p* < 0.001).

Bioimpedance analysis (BIA) showed a significant increase in both total body fat and trunk fat mass across BMI categories (*p* < 0.001). In contrast, fat-free mass was higher in the Normal weight group, declining with increasing obesity (*p* = 0.03). Similarly, muscle mass and hydration status decreased with higher obesity levels (*p* < 0.001).

Several biological markers also exhibited significant changes. Fasting blood glucose (FBG) increased from 80.00 mg/dL in the control group to 114.00 mg/dL in the Obesity III group (*p* < 0.001). Total cholesterol (TC) and LDL cholesterol (LDL-c) increased significantly (*p* < 0.001), while HDL cholesterol (HDL-c) decreased (*p* = 0.01), particularly in the Obesity III group. Triglycerides and uric acid levels did not show significant differences across BMI categories.

Liver function markers, AST and ALT, increased significantly with higher BMI (*p* < 0.001). Serum vitamin D levels progressively decreased across BMI categories (*p* < 0.001). Thyroid function markers, including TSH and FT4, did not show significant differences across BMI categories.

Detailed results are presented in Table 1, Figure 3 and Figure 4.

The analysis of cardiometabolic and anthropometric variables across age groups, classified as Young Adults (≤30 years), Early Middle Age (31–45 years), Late Middle Age (46–60 years), and Older Adults (≥61 years), reveals significant trends in metabolic health and cardiovascular risk factors. Notably, TMAO and CIMT exhibit significant increases with age, while other markers such as BMI, insulin resistance, and lipid profiles follow expected but statistically non-significant trends. BMI significantly increases with age (*p* = 0.04), rising from a median of 28.50 kg/m^2^ in Young Adults to 37.00 kg/m^2^ in Older Adults. However, fat mass and fat-free mass did not show statistically significant differences across age groups.

TMAO levels increased significantly with age (*p* = 0.02), with a steady rise from 192.00 in Young Adults to 388.00 in Older Adults. Resistin levels did not show a statistically significant increase across age groups (*p* = 0.63), though a modest rise was observed in Older Adults. CIMT increased significantly with age on both the right carotid artery (*p* = 0.02) and left carotid artery (*p* = 0.01). Right CIMT rose from 0.66 mm in Young Adults to 0.88 mm in Older Adults, while left CIMT followed a similar trend. WHR showed an increasing trend but was not statistically significant (*p* = 0.19). Metabolic markers such as FBG, HbA1c, and HOMA-IR increased with age, although none reached statistical significance. FBG rose from 92.00 mg/dL in young adults to 114.00 mg/dL in older adults. Lipid metabolism showed trends of deterioration with age. TC and LDL-C levels increased, while HDLc levels decreased, aligning with the typical age-related worsening of lipid profiles. However, these differences were not statistically significant. Liver enzyme levels (AST and ALT) tended to rise with age, particularly in Older Adults, suggesting potential age-related liver stress. Vitamin D levels remained low across all age groups, reflecting a common deficiency in both younger and older populations. The results are presented in Table 2.

The analysis of cardiometabolic and anthropometric parameters across sex categories reveals notable differences in body composition, metabolism, and biochemical markers, while TMAO, resistin, and vascular health (CIMT) did not show significant sex-based variation. One of the most pronounced sex differences was observed in body composition. Females exhibited significantly higher fat mass (median: 42.00 kg) compared to males (31.00 kg, *p* = 0.008). Trunk fat was also significantly higher in females (41.00 kg) than in males (34.00 kg, *p* = 0.05). Conversely, fat-free mass and muscle mass were significantly higher in males (*p* < 0.001 and *p* = 0.001, respectively). Resting basal metabolism (RBM) was significantly higher in males, with a median of 2193.00 kcal/day compared to 1568.00 kcal/day in females (*p* < 0.001). Similarly, body water content was markedly higher in males (51.00%) compared to females (42.00%, *p* < 0.001). Cardiovascular and vascular health parameters, including carotid intima-media thickness (CIMT), did not differ significantly between sexes (*p* = 0.49 for right CIMT and *p* = 0.60 for left CIMT). Waist–hip ratio (WHR) was comparable between men and women (*p* = 0.91). TMAO and resistin levels did not show significant differences between sexes (*p* = 0.73 and *p* = 0.33, respectively). Fasting blood glucose (FBG) and insulin resistance (HOMA-IR) were also similar between sexes (*p* = 0.34 and *p* = 0.92, respectively). Lipid profiles exhibited some sex-based trends, but none reached statistical significance. LDL cholesterol was lower in males (100.00 mg/dL) than in females (118.00 mg/dL, *p* = 0.09), while HDL cholesterol was slightly higher in males (52.00 mg/dL) than in females (48.00 mg/dL, *p* = 0.24). Triglycerides and total cholesterol levels were similar between sexes (*p* = 0.62 and *p* = 0.28, respectively). Vitamin D levels were significantly higher in males (31.00 ng/mL) compared to females (23.00 ng/mL, *p* = 0.04). Liver enzymes (AST and ALT) and thyroid markers (TSH and FT4) did not show significant sex-based differences. The results are summarized in Table 3.

This study explored associations between TMAO and resistin levels with demographic, behavioral, and health-related factors, yielding several significant findings. Gender did not have a significant influence on TMAO (*p* = 0.73) or resistin levels (*p* = 0.33), with comparable median values across males and females. Alcohol consumption and smoking status also did not significantly affect TMAO (*p* = 0.68) or resistin (*p* = 0.15). Shorter sleep duration was significantly associated with higher resistin levels (*p* = 0.04), while TMAO levels remained unchanged (*p* = 0.91). Physical activity was inversely correlated with both biomarkers, with lower levels of activity associated with significantly higher TMAO (*p* = 0.001) and resistin (*p* = 0.02). TMAO was significantly associated with a family medical history (FMH) of obesity (*p* = 0.03) and diabetes (*p* = 0.01). Resistin was significantly associated with a family history of diabetes (*p* = 0.04).

The findings are summarized in Table 4.

### 3.2. Correlation Analysis of TMAO and Resistin with Cardiometabolic Risk Factors

The analysis demonstrated significant correlations between TMAO levels and multiple metabolic health parameters. TMAO exhibited a moderate positive correlation with WHR (ρ = 0.463, *p* < 0.001) and BMI (ρ = 0.508, *p* < 0.001), reinforcing its association with both central and generalized adiposity. Additionally, correlations between TMAO and liver enzymes ALT (ρ = 0.56, *p* < 0.001) and AST (ρ = 0.480, *p* < 0.001) suggest that elevated TMAO levels may reflect underlying liver dysfunction, particularly in individuals with overweight and obesity. Furthermore, TMAO demonstrated moderate correlations with HbA1c (ρ = 0.442, *p* < 0.001) and HOMA-IR (ρ = 0.461, *p* < 0.001), supporting its potential role in impaired glycemic regulation and insulin resistance.

TMAO levels also showed moderate correlations with LDL-c (ρ = 0.428, *p* < 0.001) and TC (ρ = 0.499, *p* < 0.001), indicating a possible role in dyslipidemia and elevated cardiovascular risk. The most pronounced relationships were observed between TMAO and CIMT on both the right (ρ = 0.674, *p* < 0.001) and left (ρ = 0.680, *p* < 0.001) carotid arteries, suggesting a strong link between higher TMAO concentrations and increased carotid artery thickness, a key marker of subclinical atherosclerosis. The results are detailed in Table 5 and Figure 5.

The analysis demonstrated a statistically significant positive correlation between resistin levels and CIMT on both sides. Specifically, resistin showed a correlation of τ = 0.317 (*p* < 0.001) with left CIMT and τ = 0.336 (*p* < 0.001) with right CIMT. These findings suggest that higher resistin levels are associated with increased carotid artery thickness bilaterally, indicating a potential relationship between resistin and subclinical atherosclerosis.

### 3.3. Significant Independent Predictors of TMAO Levels

A multiple linear regression analysis was conducted to identify significant predictors of TMAO levels. The model demonstrated strong explanatory power, with an adjusted R^2^ of 0.483, indicating that 48.3% of the variance in TMAO levels is explained by the predictors included. Right CIMT, trunk fat mass, abdominal circumference, and smoking status were identified as significant predictors. Right CIMT showed a strong positive association with TMAO, with an estimate of 1090.09 (CI: 766.08–1414.10, *p* < 0.001), suggesting a direct relationship between arterial thickness and TMAO levels. Trunk fat mass was also positively associated with TMAO, with an estimate of 9.11 (CI: 2.74–15.49, *p* = 0.006), linking body composition to circulating TMAO concentrations. The results are presented in Table 6.

### 3.4. Assessing Cardiometabolic Risk Using TMAO: Logistic Regression Analysis

The logistic regression analysis assessed the predictors of high cardiometabolic risk using a TMAO threshold to classify participants. The model demonstrated strong explanatory power, with a Nagelkerke R^2^ of 0.475, accounting for 47.5% of the variance in cardiometabolic risk. Total cholesterol (TC) was identified as a significant predictor, with an odds ratio (OR) of 1.03 (95% CI: 1.01–1.05, *p* = 0.015), indicating that each unit increase in TC raises the likelihood of high cardiometabolic risk by 3%. Free thyroxine (FT4) was inversely associated with high cardiometabolic risk, with an OR of 0.72 (95% CI: 0.52–0.95, *p* = 0.029), suggesting a 28% reduction in risk per unit increase in FT4. Additionally, AST showed a positive association with high cardiometabolic risk (OR: 1.10, 95% CI: 1.03–1.20, *p* = 0.010), meaning each unit increase in AST was linked to a 10% higher risk classification. The results are presented in Table 7.

The ROC analysis evaluates the predictive performance of the logistic regression model for cardiometabolic risk based on TMAO levels. The AUC is 0.880, indicating excellent discrimination, meaning the model has an 88% likelihood of accurately distinguishing between high-risk and non-risk patients. The model achieves an accuracy of 91.7%, reflecting its strong capacity to correctly classify patients into the appropriate risk categories.

Specificity is notably high at 97.8%, demonstrating the model’s effectiveness in correctly identifying patients without high cardiometabolic risk, minimizing false positives. Sensitivity, which reflects the model’s ability to identify high-risk patients, is 71.4%. The confidence interval for the AUC (0.768–0.992) highlights the reliability of the model, with the narrow interval indicating consistent performance across different samples. The *p*-value of <0.001 confirms that the model’s discriminatory ability is statistically significant and not due to chance. Full details are presented in Figure 6.

### 3.5. Significant Independent Predictors of Resistin Levels

The multiple linear regression analysis identified significant predictors of resistin levels, explaining 44.3% of the variance (adjusted R^2^ = 0.443). Left CIMT was strongly associated with resistin, with an estimated coefficient of 10.63 (CI: 4.42–16.84, *p* = 0.001), indicating that each unit increase in left CIMT corresponds to a 10.63-unit increase in resistin levels. WHR was also positively correlated with resistin levels (estimate: 3.11, CI: 1.01–5.21, *p* = 0.004), linking central adiposity to elevated resistin concentrations. Total cholesterol (TC) showed a small but significant positive association with resistin (estimate: 0.04, CI: 0.01–0.06, *p* = 0.019), while LDL-c exhibited a significant inverse association with resistin levels (estimate: −0.06, CI: −0.09–−0.02, *p* = 0.003). Trunk fat percentage measured by bioelectrical impedance did not show a significant association with resistin levels. The results are presented in Table 8.

### 3.6. Interaction Between TMAO and Resistin in Predicting Obesity Risk

The logistic regression model evaluated the predictive value of TMAO, resistin, and their interaction in relation to obesity, explaining 32.1% of the variance (Nagelkerke R^2^ = 0.321). TMAO was a significant predictor of obesity (OR: 1.02, CI: 1.01–1.04, *p* = 0.003), with each unit increase in TMAO corresponding to a 2% higher obesity risk, supporting its role as a biomarker for metabolic health. Resistin was also significantly associated with obesity (OR: 1.93, CI: 1.19–3.39, *p* = 0.012), with each unit increase leading to a 93% increase in obesity risk, emphasizing its link to inflammation and metabolic dysfunction. The interaction term between TMAO and resistin was statistically significant (OR: 1.00, CI: 1.00–1.00, *p* = 0.010), indicating that their combined effect on obesity differs from their individual contributions. The results are presented in Table 9.

The interaction plot in Figure 7 illustrates the relationship between TMAO levels and the probability of obesity across different resistin levels. TMAO serves as the primary predictor, while resistin acts as a moderating variable. The plot displays three distinct trends: at high resistin levels (+1 SD), the probability of obesity decreases as TMAO levels rise. At average resistin levels (mean), a positive association is observed between TMAO and obesity, where increasing TMAO levels correspond to a higher obesity risk. In contrast, at low resistin levels (−1 SD), the probability of obesity rises sharply with increasing TMAO levels. These findings suggest a complex interaction between TMAO and resistin in predicting obesity risk, as shown in Figure 7.

A logistic regression model was developed to assess the risk of insulin resistance using resistin, TMAO, and their interaction as predictors. The model explains 36.5% of the variance in insulin resistance risk (Nagelkerke R^2^ = 0.365), indicating a moderately strong predictive ability. Resistin was a significant predictor of insulin resistance, with an OR of 2.85 (CI: 1.57–5.98, *p* = 0.002), meaning that each unit increase in resistin nearly triples the odds of insulin resistance. TMAO was also a significant predictor, with an OR of 1.03 (CI: 1.01–1.05, *p* = 0.005), indicating that each unit increase in TMAO raises the odds of insulin resistance by 3%. The interaction between resistin and TMAO was statistically significant (OR: 1.00, CI: 0.99–1.00, *p* = 0.004), suggesting that the effects of resistin on insulin resistance may be modified by TMAO levels. The results highlight a complex interplay between these biomarkers in predicting insulin resistance. The results are presented in Table 10.

## 4. Discussion

The present study examined the associations between TMAO and resistin levels with demographic, behavioral, and health-related factors, identifying key relationships relevant to cardiometabolic risk. While previous research, such as the study by Zhuang et al., suggested higher TMAO levels in males due to differences in gut microbiota and dietary patterns [70], our findings did not confirm a significant sex-based variation for either TMAO or resistin. Additionally, elevated TMAO levels have been linked to a family history of obesity and diabetes, suggesting a genetic or environmental predisposition influencing its metabolism [71]. Consistent with this, our study observed significantly higher TMAO and resistin levels in participants with a family history of obesity, diabetes, or cardiovascular disease, reinforcing the potential hereditary influence on these biomarkers. These findings underscore the need for further research to elucidate the mechanisms underlying familial and metabolic influences on TMAO and resistin.

Smoking has been associated with elevated TMAO and resistin levels, likely due to gut microbiota alterations and increased oxidative stress, contributing to cardiovascular risk [72]. However, our study found no significant effect of smoking on these biomarkers. Similarly, while alcohol consumption has been linked to increased TMAO and resistin levels through inflammatory pathways and liver stress [73], no such association was observed in our study. Shorter sleep duration has been reported to elevate resistin and TMAO levels, potentially through metabolic and inflammatory changes [74]. In our study, shorter sleep correlated with higher resistin but did not affect TMAO levels.

This study provides a comprehensive analysis of the associations between resistin, TMAO, and subclinical atherosclerosis, measured through CIMT. It also identifies independent predictors of these biomarkers in individuals with overweight and obesity, offering insights into their potential role in early cardiometabolic dysfunction.

Obesity is a major risk factor for insulin resistance, which contributes to subclinical atherosclerosis [75]. Endothelial dysfunction, driven by oxidative stress, inflammation, hypertension, and aging, leads to arterial narrowing and vascular impairment. While overweight individuals may exhibit vascular dysfunction, its severity varies. Aging-related vascular changes, including collagen deposition, smooth muscle degradation, and arterial stiffening, further compromise elasticity [76]. Subclinical atherosclerosis, detected through CIMT measurement, is significantly elevated in obesity and insulin resistance [36]. CIMT serves as an early marker of arterial wall changes, allowing for risk assessment before clinical symptoms appear [36]. It is strongly associated with cardiovascular risk factors such as hypertension and dyslipidemia, reflecting their cumulative effect on vascular health [77]. In terms of cardiovascular health, the current study identified strong correlations between both right and left CIMT and serum TMAO levels, indicating an elevated likelihood of subclinical atherosclerosis in patients with obesity who exhibit higher TMAO levels. Moreover, right CIMT was identified as a highly significant predictor of TMAO, underscoring a robust association between carotid arterial wall thickness and this biomarker. These findings suggest an enhanced role for TMAO in cardiovascular risk assessment. Additionally, moderate correlations were observed between TMAO and both LDL-c and TC, suggesting a contributory role for TMAO in dyslipidemia development, and thereby, a potential amplification in the risk of cardiovascular pathologies. From the perspective of evaluating insulin resistance through the determination of serum resistin levels, bilateral CIMT demonstrated a positive correlation with elevated serum resistin concentrations. Considering that CIMT is a well-established marker of subclinical atherosclerosis and, by extension, an indicator of cardiovascular risk, these findings suggest that resistin may play a significant role in cardiovascular pathology. Furthermore, resistin could potentially serve as a biomarker for cardiovascular disease risk assessment.

Elevated TMAO levels have been linked to adiposity, insulin resistance, and atherosclerosis through inflammatory signaling, impaired lipid metabolism, and vascular dysfunction [47,78]. Its correlation with CIMT suggests a role in early atherosclerosis, making it a potential biomarker for cardiometabolic risk. In our study, TMAO was associated with markers of obesity, insulin resistance, subclinical atherosclerosis, and liver function. Moderate correlations were observed with BMI, waist circumference, trunk fat percentage, HOMA-IR, and HbA1c, reinforcing its relevance in metabolic dysfunction. Additionally, TMAO correlated with total and LDL cholesterol, supporting its cardiovascular significance. Waist circumference, trunk fat percentage, CIMT, and smoking status emerged as key predictors of TMAO levels, highlighting their importance in clinical assessments of obesity-related risks. The significant increase in TMAO and CIMT with age highlights their potential role as markers of age-related cardiovascular risk. The observed trends in BMI, lipid profiles, and insulin resistance, despite not reaching statistical significance, suggest progressive metabolic decline with aging. The lack of significant differences in resistin levels may indicate that its role in aging-related metabolic dysfunction is more complex. The increase in liver enzymes and persistently low vitamin D levels emphasize the need for regular monitoring of metabolic and hepatic health in aging individuals to mitigate potential long-term risks.

Resistin, primarily secreted by macrophages in adipose tissue, plays a key role in obesity-related inflammation and cardiovascular disease. It contributes to adipose tissue dysfunction by promoting chronic low-grade inflammation and interfering with insulin signaling, exacerbating metabolic dysfunction [79]. In our study, resistin was significantly associated with CIMT, particularly on the left carotid side, reinforcing its role in early vascular changes linked to cardiovascular disease. A positive correlation with WHR highlights its relevance in central obesity and insulin resistance. The observed negative association between resistin and LDL cholesterol, despite its positive correlation with total cholesterol, may indicate metabolic variations within the study cohort, though this finding warrants further investigation. Additionally, our results suggest that TMAO and resistin may act synergistically, amplifying inflammatory pathways and contributing to vascular dysfunction. Both biomarkers were independently associated with insulin resistance and early atherosclerosis, supporting their role in cardiometabolic risk assessment. Their correlation with CIMT further underscores their potential as early indicators of vascular impairment in individuals with excess weight. These findings emphasize the importance of TMAO and resistin as complementary biomarkers in obesity-related metabolic and cardiovascular dysfunction, offering potential value in early risk stratification and targeted intervention strategies.

Both TMAO and resistin were identified as significant predictors of obesity, with a statistically significant interaction between them, suggesting a more complex relationship beyond their individual effects. Their interplay indicates a potential synergistic influence on obesity risk, emphasizing the importance of assessing them together for better risk prediction. The findings highlight the need for further research into the mechanistic pathways linking these biomarkers to obesity and cardiometabolic dysfunction.

Our study validated the primary hypothesis, confirming the associations between TMAO, resistin, and cardiometabolic pathology, including their links to CIMT, metabolic markers, and body composition parameters. Combining these biomarkers improved obesity risk prediction, while additional correlations with anthropometric and bioimpedance measures provided further insights into their metabolic significance. Despite these findings, limitations remain in assessing TMAO, resistin, and CIMT in individuals with obesity, warranting further investigation. First, the small sample size limits statistical power, particularly in subgroup analyses based on BMI categories, age groups, sex, and TMAO thresholds. This may lead to inflated effect sizes and reduce the generalizability of our findings. Also, the cross-sectional nature of data collection limits our ability to draw causal conclusions. The associations observed in this study should therefore be interpreted with caution. Future studies with larger cohorts and longitudinal designs are necessary to validate these findings and establish causality. Another limitation of this study is the strict definition of the control group, which excluded participants with common cardiometabolic risk factors. While this approach ensured a clearer distinction between groups, it may limit the generalizability of our findings to a broader population. Future studies with larger sample sizes could include a more heterogeneous control group and apply statistical adjustments for common metabolic risk factors to further assess the generalizability of these findings. CIMT measurements require the use of advanced technology and specific equipment, as well as strict adherence to procedural guidelines, including appropriate conditions and the involvement of a certified clinician with expertise in this field. Moreover, the determination of TMAO and resistin in laboratory settings involves additional costs, which may pose challenges for broader application. Nonetheless, this study highlights the critical role of these biomarkers in predicting excessive weight and underscores the potential for their combined use to enhance risk stratification and clinical decision-making in obesity management.

## 5. Conclusions

This study underscores the role of TMAO as a key metabolic mediator linking increased BMI with metabolic dysfunction and cardiovascular disease. The strong association between circulating TMAO levels and central adiposity highlights its involvement in obesity-related metabolic disturbances. Furthermore, its correlation with CIMT, a validated marker of subclinical atherosclerosis, supports its contribution to atherosclerotic progression. Resistin also emerged as a relevant biomarker associated with cardiometabolic risk factors, including dyslipidemia, inflammation, and vascular remodeling. The interaction between TMAO and resistin suggests a collective influence on obesity-related insulin resistance and cardiovascular complications, reinforcing their importance in metabolic risk assessment. Additionally, this study examined variations across age groups and sexes, offering a deeper understanding of how these biomarkers relate to metabolic health and vascular integrity in diverse populations. The findings suggest that TMAO and resistin may serve as valuable biomarkers for identifying individuals at higher risk of cardiometabolic complications, with potential applications in risk stratification and early intervention. These results contribute to the growing evidence supporting the role of gut-derived metabolites and adipokines in metabolic and vascular health. Integrating TMAO and resistin into routine assessments could enhance personalized risk evaluation and aid in developing targeted therapeutic strategies to mitigate obesity-related metabolic and cardiovascular complications.

## Figures and Tables

**Figure 1 nutrients-17-00798-f001:**
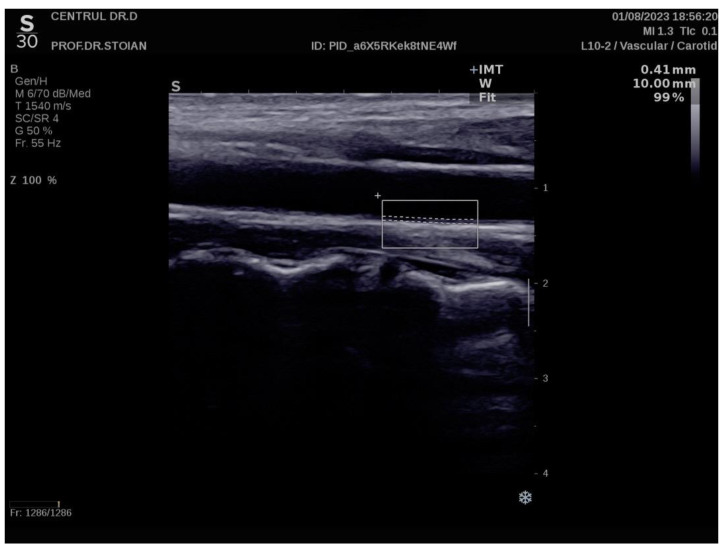
Example of a normal left CIMT measurement in a normoweight patient using the Aixplorer MACH 30 ultrasound system.

**Figure 2 nutrients-17-00798-f002:**
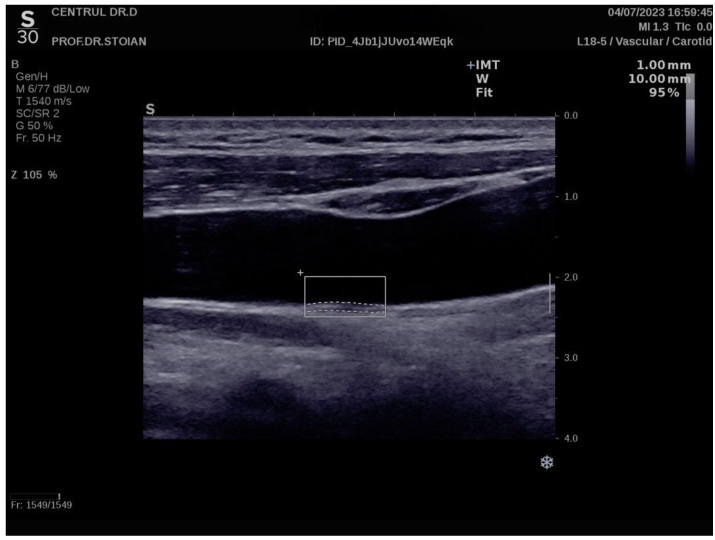
Example of a high value of left CIMT measurement in a subject with obesity grade II using the Aixplorer MACH 30 ultrasound system.

**Figure 3 nutrients-17-00798-f003:**
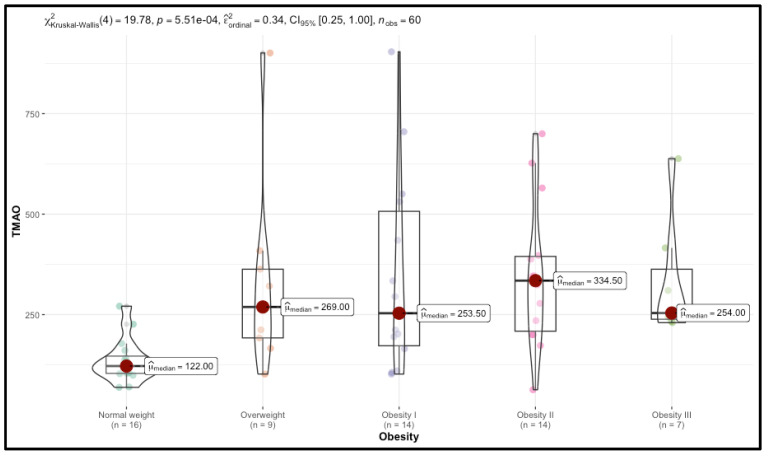
Distribution of TMAO levels across different BMI categories.

**Figure 4 nutrients-17-00798-f004:**
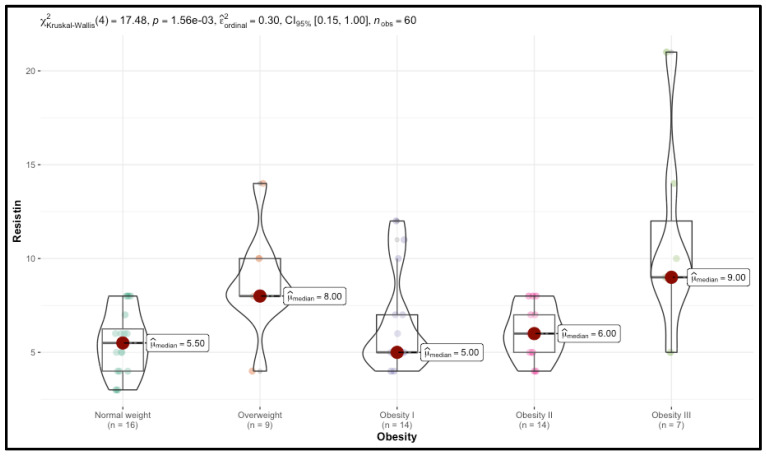
Distribution of resistin levels across different BMI categories.

**Figure 5 nutrients-17-00798-f005:**
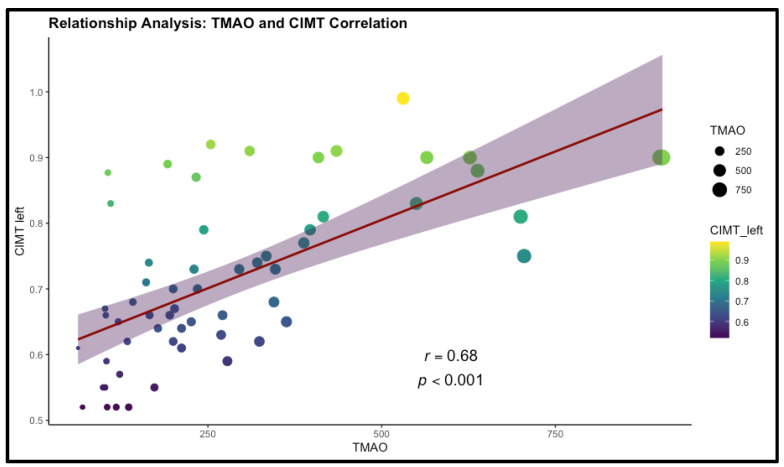
Correlation between TMAO levels and left carotid intima-media thickness (CIMT).

**Figure 6 nutrients-17-00798-f006:**
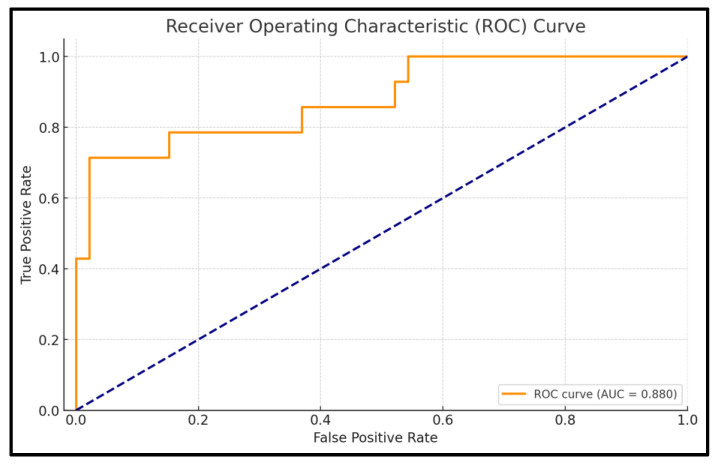
ROC curve for TMAO-based cardiometabolic risk prediction model.

**Figure 7 nutrients-17-00798-f007:**
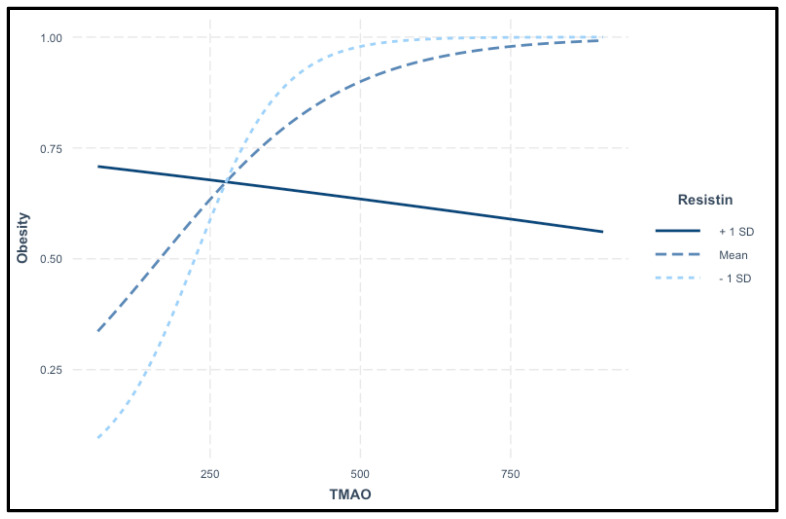
Interaction of TMAO and resistin in predicting obesity.

**Table 1 nutrients-17-00798-t001:** Comparison of demographic, metabolic, and anthropometric variables across BMI categories.

Variable	Normal (*N* = 16)	Overweight (*N* = 9)	Obesity I (*N* = 14)	Obesity II (*N* = 14)	Obesity III (*N* = 7)	*p*-Value
**Age**	28.00 (23.75–33.50)	28.00 (24.00–32.00)	33.50 (26.00–44.75)	41.00 (26.00–54.50)	51.00 (35.50–62.00)	0.19
**TMAO**	122.00 (103.75–146.75)	269.00 (192.00–363.00)	253.50 (172.50–545.25)	334.50 (208.75–394.75)	254.00 (238.50–363.00)	<0.001
**Resistin**	5.50 (4.00–6.25)	8.00 (8.00–10.00)	5.00 (5.00–7.00)	6.00 (5.00–7.00)	9.00 (9.00–12.00)	0.001
**CIMT right**	0.60 (0.56–0.68)	0.72 (0.71–0.84)	0.79 (0.71–0.93)	0.73 (0.65–0.81)	0.88 (0.85–0.91)	<0.001
**CIMT left**	0.60 (0.52–0.65)	0.67 (0.65–0.89)	0.75 (0.69–0.87)	0.70 (0.62–0.79)	0.87 (0.80–0.90)	<0.001
**WC**	87.00 (77.75–89.00)	91.00 (88.00–96.00)	105.50 (98.25–114.25)	120.50 (116.25–123.00)	136.00 (130.50–146.50)	<0.001
**WHR**	0.83 (0.82–0.86)	0.87 (0.87–0.90)	0.97 (0.90–1.10)	1.23 (1.12–1.40)	1.80 (1.70–1.90)	<0.001
**Fat mass**	29.50 (26.75–37.25)	38.50 (37.00–40.00)	42.00 (39.55–43.75)	46.75 (44.00–49.50)	47.00 (42.55–49.50)	<0.001
**Fat free mass**	59.00 (46.00–67.00)	45.00 (44.00–51.00)	51.00 (48.50–53.75)	52.00 (50.00–54.00)	58.00 (55.00–68.00)	0.03
**Trunk fat**	25.50 (22.75–35.25)	38.00 (34.00–39.00)	41.00 (38.25–42.00)	45.00 (41.25–48.75)	43.00 (40.50–47.00)	<0.001
**Muscle mass**	70.00 (60.00–74.00)	58.00 (57.00–60.00)	55.00 (54.00–57.75)	50.00 (47.25–53.75)	51.00 (48.50–54.50)	<0.001
**RBM**	1591.00 (1456.75–2026.25)	1383.00 (1361.00–1568.00)	1638.00 (1535.00–1685.25)	1644.50 (1543.00–1764.00)	1874.00 (1728.00–2101.00)	0.03
**Body water**	55.50 (45.83–60.25)	45.00 (44.00–46.00)	43.00 (41.00–44.00)	39.00 (38.00–41.00)	39.00 (37.10–42.00)	<0.001
**FBG**	80.00 (76.75–90.50)	91.00 (87.00–100.00)	95.00 (89.00–107.50)	101.50 (98.50–110.00)	114.00 (103.50–150.00)	<0.001
**HbA1c**	5.00 (4.80–5.15)	5.40 (5.40–5.70)	5.65 (5.40–6.25)	5.75 (5.53–5.88)	6.00 (5.75–7.00)	<0.001
**HOMA-IR**	1.85 (1.35–2.23)	2.20 (2.00–3.00)	3.75 (2.42–4.88)	4.85 (4.60–5.85)	8.00 (4.60–10.30)	<0.001
**TC**	139.00 (111.75–153.25)	210.00 (182.00–220.00)	208.00 (175.50–241.50)	195.00 (147.75–206.00)	212.00 (173.50–238.50)	<0.001
**LDLc**	84.00 (76.75–91.25)	132.00 (96.00–153.00)	138.00 (102.75–176.50)	122.50 (97.75–137.50)	115.00 (111.00–133.50)	<0.001
**HDLc**	57.00 (51.00–61.00)	49.00 (44.00–58.00)	49.00 (43.00–58.00)	46.50 (37.50–51.25)	38.00 (35.00–46.50)	0.01
**Triglycerides**	90.00 (83.50–102.25)	121.00 (77.00–190.00)	99.00 (79.00–145.75)	107.00 (91.75–171.00)	109.00 (102.00–192.50)	0.23
**Uric acid**	4.00 (3.00–6.00)	5.00 (4.00–6.00)	5.00 (4.10–6.00)	5.00 (5.00–6.00)	6.00 (4.70–6.80)	0.25
**Vitamin D**	31.50 (23.75–34.25)	27.00 (24.00–29.00)	22.00 (19.25–24.00)	21.00 (18.00–24.25)	19.00 (13.80–23.00)	<0.001
**TSH**	2.90 (2.08–3.40)	1.70 (1.10–2.40)	2.70 (2.01–3.45)	2.40 (1.72–3.40)	3.00 (2.10–4.00)	0.28
**FT4**	10.50 (8.93–14.00)	11.00 (9.30–12.40)	13.40 (11.00–14.38)	11.50 (9.48–13.23)	15.00 (13.00–15.50)	0.07
**AST**	14.50 (11.00–20.25)	28.00 (21.00–48.00)	27.00 (21.00–35.75)	25.50 (20.75–35.00)	30.00 (23.50–35.00)	<0.001
**ALT**	14.00 (12.00–20.25)	26.00 (22.00–55.00)	32.50 (24.00–42.50)	28.00 (23.50–37.50)	29.00 (21.50–44.00)	<0.001

Abbreviations: TMAO—trimethylamine N-oxide, CIMT—carotid intima-media thickness, WC—waist circumference, WHR—waist–hip ratio, RBM—resting basal metabolism, FBG—fasting blood glucose, HOMA-IR—homeostasis model assessment of insulin resistance, LDLc—low-density lipoprotein cholesterol, HDLc—high-density lipoprotein cholesterol, TSH—thyroid-stimulating hormone, FT4—free thyroxine (free T4), AST—aspartate aminotransferase, ALT—alanine aminotransferase, TC—total cholesterol, *p*-value—Kruskal–Wallis test result, N—number of individuals.

**Table 2 nutrients-17-00798-t002:** Comparison of demographic, metabolic, and anthropometric variables across age categories.

Variable	Young Adults (*N* = 27)	Early Middle Age (*N* = 16)	Late Middle Age (*N* = 12)	Older Adults (*N* = 5)	*p*-Value
**BMI**	28.50 (24.50–33.50)	31.70 (27.65–36.97)	34.50 (28.40–37.38)	37.00 (37.00–41.60)	0.04
**TMAO**	192.00 (105.50–273.50)	234.00 (165.25–409.75)	294.50 (165.75–439.00)	388.00 (295.00–416.00)	0.02
**Resistin**	6.00 (5.00–8.00)	5.50 (4.75–7.25)	6.50 (5.00–8.00)	8.00 (6.00–9.00)	0.63
**CIMT right**	0.66 (0.58–0.80)	0.72 (0.69–0.80)	0.78 (0.74–0.85)	0.88 (0.81–0.91)	0.02
**CIMT left**	0.63 (0.57–0.78)	0.69 (0.66–0.77)	0.73 (0.70–0.82)	0.81 (0.77–0.90)	0.01
**WC**	105.00 (89.00–114.50)	96.50 (90.50–118.25)	118.50 (87.50–123.25)	113.00 (110.00–114.00)	0.48
**WHR**	0.93 (0.86–0.97)	1.03 (0.85–1.25)	1.22 (0.88–1.42)	1.40 (0.96–1.60)	0.19
**Fat mass**	41.00 (37.50–43.50)	39.20 (31.00–46.62)	43.50 (35.50–47.00)	44.00 (37.00–45.00)	0.78
**Fat free mass**	52.00 (48.00–55.00)	53.50 (45.75–64.75)	51.50 (47.25–58.25)	50.00 (50.00–54.00)	0.89
**Trunk fat**	40.00 (35.50–42.50)	37.00 (32.00–45.25)	40.00 (36.00–43.50)	40.00 (39.00–42.00)	0.77
**Muscle mass**	57.00 (55.00–60.00)	58.00 (50.00–67.00)	54.00 (50.75–62.00)	53.00 (53.00–60.00)	0.69
**RBM**	1631.00 (1540.50–1736.00)	1645.00 (1416.75–2164.25)	1607.50 (1467.25–1809.75)	1630.00 (1567.00–1704.00)	0.96
**Body water**	43.00 (42.00–46.00)	44.50 (38.88–52.25)	41.00 (39.00–44.75)	41.00 (40.00–44.00)	0.54
**FBG**	92.00 (85.50–100.00)	97.50 (87.75–105.75)	99.00 (90.00–120.00)	114.00 (100.00–154.00)	0.13
**HbA1c**	5.50 (5.20–5.80)	5.35 (5.10–5.62)	5.75 (5.35–6.12)	6.00 (5.80–6.40)	0.09
**HOMA-IR**	3.00 (1.90–4.05)	3.10 (2.00–5.00)	3.50 (2.08–6.25)	5.00 (4.30–5.00)	0.20
**TC**	160.00 (145.00–209.50)	174.00 (129.00–211.00)	192.00 (161.00–220.75)	212.00 (206.00–214.00)	0.18
**LDLc**	100.00 (83.00–127.50)	114.50 (85.50–147.00)	121.00 (99.00–140.00)	123.00 (122.00–135.00)	0.45
**HDLc**	48.00 (43.50–61.00)	54.00 (48.25–57.25)	45.00 (38.00–53.50)	48.00 (43.00–49.00)	0.42
**Triglycerides**	93.00 (72.50–132.50)	98.00 (88.00–128.00)	112.50 (87.50–188.25)	120.00 (110.00–211.00)	0.15
**Uric acid**	5.00 (4.00–6.00)	5.00 (3.95–6.00)	5.00 (4.75–6.25)	7.00 (4.40–7.00)	0.63
**Vitamin D**	23.00 (19.50–29.50)	23.00 (17.25–32.00)	25.00 (21.75–30.00)	22.00 (19.00–24.00)	0.40
**TSH**	2.60 (1.70–3.40)	2.20 (1.75–2.85)	2.70 (2.30–3.65)	2.80 (1.30–3.00)	0.28
**FT4**	13.00 (10.65–14.00)	13.15 (9.52–15.00)	11.00 (9.75–12.75)	11.00 (9.00–12.00)	0.54
**AST**	21.00 (16.50–26.00)	22.50 (17.00–30.00)	30.00 (21.00–36.25)	35.00 (30.00–36.00)	0.16
**ALT**	22.00 (14.50–26.50)	25.50 (16.75–31.00)	37.00 (22.75–41.75)	38.00 (34.00–48.00)	0.10

Abbreviations: TMAO—trimethylamine N-oxide, CIMT—carotid intima-media thickness, WC—waist circumference, WHR—waist–hip ratio, RBM—resting basal metabolism, FBG—fasting blood glucose, HOMA-IR—homeostasis model assessment of insulin resistance, LDLc—low-density lipoprotein cholesterol, HDLc—high-density lipoprotein cholesterol, TSH—thyroid-stimulating hormone, FT4—free thyroxine (Free T4), AST—aspartate aminotransferase, ALT—alanine aminotransferase, TC—total cholesterol, *p*-value—Kruskal–Wallis test result, N—number of individuals.

**Table 3 nutrients-17-00798-t003:** Comparison of demographic, metabolic, and anthropometric variables across sex categories.

Variable	Female (*N* = 47)	Male (*N* = 13)	*p*-Value
**Age**	31.00 (24.50–51.00)	39.00 (29.00–51.00)	0.28
**BMI**	32.50 (27.75–36.90)	28.10 (23.50–37.00)	0.32
**TMAO**	226.00 (150.50–355.00)	233.00 (123.00–321.00)	0.73
**Resistin**	6.00 (5.00–8.00)	8.00 (5.00–9.00)	0.33
**CIMT right**	0.72 (0.65–0.83)	0.77 (0.69–0.89)	0.49
**CIMT left**	0.67 (0.62–0.80)	0.74 (0.66–0.83)	0.60
**WC**	106.00 (90.00–118.50)	105.00 (89.00–118.00)	0.79
**WHR**	0.95 (0.87–1.23)	0.95 (0.84–1.40)	0.91
**Fat mass**	42.00 (38.00–46.30)	31.00 (29.00–39.00)	0.008
**Fat free mass**	50.00 (46.00–54.00)	67.00 (58.00–74.00)	<0.001
**Trunk fat**	41.00 (37.00–44.00)	34.00 (25.00–41.00)	0.05
**Muscle mass**	55.00 (51.00–59.00)	66.00 (60.00–71.00)	0.001
**RBM**	1568.00 (1453.50–1680.50)	2193.00 (1988.00–2377.00)	<0.001
**Body water**	42.00 (39.00–45.00)	51.00 (46.00–61.00)	<0.001
**FBG**	97.00 (88.00–109.00)	89.00 (80.00–100.00)	0.34
**HbA1c**	5.60 (5.30–5.85)	5.40 (5.00–5.80)	0.48
**HOMA-IR**	3.00 (2.00–4.85)	3.00 (1.90–6.00)	0.92
**TC**	190.00 (146.00–218.00)	174.00 (145.00–190.00)	0.28
**LDLc**	118.00 (91.50–143.00)	100.00 (80.00–111.00)	0.09
**HDLc**	48.00 (41.50–60.00)	52.00 (48.00–56.00)	0.24
**Triglycerides**	104.00 (80.50–156.00)	99.00 (91.00–103.00)	0.62
**Uric acid**	5.00 (4.00–6.00)	6.00 (4.00–7.00)	0.29
**Vitamin D**	23.00 (19.00–27.00)	31.00 (22.00–32.00)	0.04
**TSH**	2.60 (1.70–3.40)	2.40 (2.00–4.00)	0.75
**FT4**	12.00 (10.00–14.00)	11.00 (9.00–15.00)	0.53
**AST**	23.00 (19.00–31.50)	21.00 (14.00–35.00)	0.69
**ALT**	25.00 (18.00–35.00)	23.00 (15.00–39.00)	0.94

Abbreviations: TMAO—trimethylamine N-oxide, CIMT—carotid intima-media thickness, WC—waist circumference, WHR—waist–hip ratio, RBM—resting basal metabolism, FBG—fasting blood glucose, HOMA-IR—homeostasis model assessment of insulin resistance, LDLc—low-density lipoprotein cholesterol, HDLc—high-density lipoprotein cholesterol, TSH—thyroid-stimulating hormone, FT4—free thyroxine (free T4), AST—aspartate aminotransferase, ALT—alanine aminotransferase, TC—total cholesterol, *p*-value—Kruskal–Wallis test result, N—number of individuals.

**Table 4 nutrients-17-00798-t004:** Associations of TMAO and resistin levels with demographic, behavioral, and health variables.

Variable	Class (N)	TMAO	*p*-Value	Resistin	*p*-Value
**Sex**	M (13)	233.00 (123.00–321.00)	0.73	8.00 (5.00–9.00)	0.33
	F (47)	226.00 (150.50–355.00)		6.00 (5.00–8.00)	
**Smoker**	Yes (17)	200.00 (110.00–363.00)	0.68	8.00 (6.00–8.00)	0.15
	No (43)	230.00 (153.50–346.00)		6.00 (5.00–8.00)	
**FMH obesity**	Yes (30)	286.50 (200.50–406.00)	0.03	7.00 (5.00–8.00)	0.12
	No (30)	172.00 (118.75–270.50)		6.00 (5.00–8.00)	
**FMH diabetes**	Yes (22)	302.50 (209.75–414.25)	0.01	7.50 (6.00–9.75)	0.04
	No (38)	193.50 (118.75–318.25)		6.00 (5.00–8.00)	
**FMH CV**	Yes (24)	286.50 (198.75–410.75)	0.02	6.00 (5.00–8.25)	0.71
	No (36)	196.00 (116.00–316.00)		6.00 (5.00–8.00)	
**Alcohol**	Yes (7)	200.00 (106.50–348.50)	0.61	6.00 (5.00–9.50)	0.89
	No (53)	230.00 (142.00–347.00)		6.00 (5.00–8.00)	
**Sleep< 7 h/night**	Yes (28)	229.50 (132.75–351.00)	0.91	7.00 (5.75–10.00)	0.04
	No (32)	221.00 (140.00–355.75)		5.50 (5.00–8.00)	
**PA < 150 min/week**	Yes (40)	273.50 (194.25–400.00)	0.001	7.50 (5.00–9.00)	0.02
	No (20)	138.00 (104.50–215.50)		5.50 (4.75–6.25)	
**Lipid profile modifications**	Yes (33)	310.00 (202.00–416.00)	0.001	6.00 (5.00–8.00)	0.67
	No (27)	161.00 (114.00–234.00)		6.00 (5.00–8.00)	
**Diabetes status**	Diabetes (9)	269.00 (192.00–324.00)	0.04	8.00 (6.00–10.00)	0.26
	Prediabetes (11)	388.00 (269.50–579.00)		6.00 (5.00–8.00)	
	Normal (40)	200.00 (122.50–288.75)		6.00 (5.00–8.00)	
**Insulin resistance**	Yes (47)	244.00 (175.50–392.50)	0.01	7.00 (5.00–8.00)	0.05
	No (13)	121.00 (105.00–212.00)		5.00 (4.00–6.00)	
**Vitamin D status**	Deficit (17)	278.00 (200.00–363.00)	0.08	6.00 (5.00–10.00)	0.30
	Insufficiency (27)	230.00 (172.00–416.00)		7.00 (5.00–8.00)	
	Optimal (16)	128.50 (108.75–244.00)		5.50 (4.00–7.25)	
**Right vascular impairment**	Yes (48)	261.50 (194.25–400.00)	<0.001	7.00 (5.00–8.25)	0.07
	No (12)	104.50 (91.75–134.50)		5.50 (4.75–6.00)	
**Left vascular impairment**	Yes (49)	254.00 (192.00–397.00)	<0.001	7.00 (5.00–8.00)	0.16
	No (11)	105.00 (100.50–129.50)		6.00 (5.00–6.00)	

Abbreviations: FMH—family medical history, CV—cardiovascular, PA—physical activity, TMAO—trimethylamine N-oxide, *p*-value—Mann–Whitney U test result, N—number of individuals.

**Table 5 nutrients-17-00798-t005:** Spearman’s correlation and effect size of TMAO levels with various health indicators.

Variable	Spearman (ρ)	*p*-Value
**WHR**	0.463	<0.001
**HBA1c**	0.442	<0.001
**HOMA-IR**	0.461	<0.001
**LDL-c**	0.428	<0.001
**CIMT right**	0.674	<0.001
**CIMT left**	0.680	<0.001
**BMI**	0.508	<0.001
**TC**	0.499	<0.001
**ALT**	0.56	<0.001
**AST**	0.480	<0.001

Abbreviations: TMAO—trimethylamine N-oxide, CIMT—carotid intima-media thickness, WHR—waist–hip ratio, HOMA-IR—homeostasis model assessment of insulin resistance, LDL-c—low-density lipoprotein cholesterol, AST—aspartate aminotransferase, ALT—alanine aminotransferase, BMI—body mass index, *p*-value—Spearman’s rank-order correlation test result.

**Table 6 nutrients-17-00798-t006:** Significant independent predictors of TMAO levels.

Predictors	Estimates	CI	*p*-Value
**CIMT right**	1090.09	766.08–1414.10	<0.001
**WC**	−4.35	−7.29–−1.40	0.005
**Trunk fat**	9.11	2.74–15.49	0.006
**Smoker (Yes)**	−94.55	−179.88–−9.23	0.031
**R^2^ adjusted**	0.483		

Abbreviations: CIMT—carotid intima-media thickness, WC—abdominal circumference, CI—95% confidence interval, *p*-value—*t*-test’s result.

**Table 7 nutrients-17-00798-t007:** Assessing cardiometabolic risk using TMAO.

Predictors	Odds Ratios	CI	*p*-Value
**TC**	1.03	1.01–1.05	0.015
**FT4**	0.72	0.52–0.95	0.029
**AST**	1.10	1.03–1.20	0.010
**R^2^ Nagelkerke**	0.475		

Abbreviations: TMAO—trimethylamine N-oxide, FT4—free thyroxine (Free T4), AST—aspartate aminotransferase, CI—95% confidence interval, *p*-value—Wald’s test result, TC—total cholesterol.

**Table 8 nutrients-17-00798-t008:** Significant independent predictors of resistin levels.

Predictors	Estimates	CI	*p*-Value
**Age**	−0.08	−0.13–−0.04	<0.001
**CIMT left**	10.63	4.42–16.84	0.001
**WHR**	3.11	1.01–5.21	0.004
**TC**	0.04	0.01–0.06	0.019
**LDL-c**	−0.06	−0.09–−0.02	0.003
**R^2^ adjusted**	0.443		

Abbreviations: CIMT—carotid intima-media thickness, WHR—waist–hip ratio, LDL-c—low-density lipoprotein cholesterol, TC—total cholesterol, CI—95% confidence interval, *p*-value—*t*-test’s result.

**Table 9 nutrients-17-00798-t009:** Interaction between TMAO and resistin for predicting obesity risk.

Predictors	Odds Ratios	CI	*p*-Value
**TMAO**	1.02	1.01–1.04	0.003
**Resistin**	1.93	1.19–3.39	0.012
**TMAO × resistin**	1.00	1.00–1.00	0.010
**R^2^ Nagelkerke**	0.321		

Abbreviations: TMAO—trimethylamine N-oxide, CI—95% confidence interval, *p*-value—Wald’s test result.

**Table 10 nutrients-17-00798-t010:** Interaction of resistin and TMAO in predicting insulin resistance.

Predictors	Odds Ratios	CI	*p*-Value
**Resistin**	2.85	1.57–5.98	0.002
**TMAO**	1.03	1.01–1.05	0.005
**Resistin × TMAO**	1.00	0.99–1.00	0.004
**R^2^ Nagelkerke**	0.365		

Abbreviations: TMAO—trimethylamine N-oxide, CI—95% confidence interval, *p*-value—Wald’s test result.

## Data Availability

The original contributions presented in the study are included in the article, further inquiries can be directed to the corresponding author.

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
