# Peer review of "Independent Predictors of Circulating Trimethylamine N-Oxide (TMAO) and Resistin Levels in Subjects with Obesity: Associations with Carotid Intima-Media Thickness and Metabolic Parameters"

_nutrients, 2025, doi:10.3390/nu17050798_

Round 1
Reviewer 1 Report
Comments and Suggestions for Authors
Review of Manuscript ID: nutrients-3481944
Title: Independent Predictors of Circulating Trimethylamine N-oxide (TMAO) and Resistin Levels in Subjects with Obesity: Associations with Carotid Intima-Media Thickness and Metabolic Parameters
Authors: Denisa Pescari, Simina Mihuta, Andreea Bena, Dana Stoian
General comment: the study reports a clinical assay conducted to examine and support the associations between serum Trimethylamine N-oxide (TMAO) and resistin levels and a range of demographic, behavioral, and health-related factors. The findings suggest that elevated TMAO levels are strongly associated with key metabolic dysregulations, including impaired glucose metabolism and insulin resistance, central features of metabolic syndrome and type 2 diabetes; besides, data indicate that higher TMAO levels are closely linked to increased carotid artery thickness, a key indicator of subclinical atherosclerosis. Moreover, results also suggest a potential link between resistin and subclinical atherosclerosis and, interestingly, there is a close relationship between both biomarkers. The authors conclude that TMAO may serve as the primary predictor for cardiometabolic dysfunction, while resistin acts as a moderating variable. The hypothesis and main objective are sound but not novel since there are already reviews and meta-analysis on the topic; however, the experimental design and cohort size are relevant and the study is rather comprehensive and well organized. Most data obtained seem to support the suggestions and conclusions of the authors, although information exposed is repetitive in results, discussion and conclusions making the manuscript rather long. Some specific comments are detailed below.
Specific comments:
1) Lines 48-49; it cannot be stated that over 1 billion individuals worldwide will be affected by obesity as of 2022 when we are already in 2025.
2) Line 58; it should say 8-11.
3) Introduction is a comprehensive review of interrelationship among carotid intima-media thickness (CIMT), metabolic parameters and cardiovascular complications but extraordinary long, comprising over 60 references. It should be shortened and some old references could be removed. The authors should consider that the exposure of the actual objective of the study starts on line 163 and at reference 46, with the introduction of the metabolite TMAO.
4) Line 176; reference 50 seems more appropriate than reference 49 to support implication of TMAO on metabolic syndrome and diabetes, please check this point.
5) Lines 246 and 249; micrograms should be noted with the Greek letter for micro and not as mc. If the authors decide to use mc, they should include this unusual acronym in abbreviations.
6) Line 271; it should be mL, capital L.
7) Page 9; it is highly unusual to start the report at figures 6 and 7 (methods) and then back to figure 1 in results. Perhaps methods should be moved down after results and discussion.
8) Since the manuscript is rather long and there are too many data presentations, perhaps tables 4-10 containing smaller information could be pooled or some of them removed and data simply exposed within the text?
9) The number of references, 113, seems more appropriate for a review than a regular research article. Indeed, the whole reference list should be double checked since there are references repeated, such as 47 and 99, as well as 51 and 101.
Author Response
Dear Esteemed Reviewer,
Thank you for your thorough review and valuable feedback. All responses to your comments can be found in the attached file, and the corresponding modifications have been incorporated into the revised manuscript submitted for your consideration. We appreciate your insightful suggestions, which have greatly contributed to improving the clarity and scientific rigor of our study.
Best regards,
The Authors

Reviewer 2 Report
Comments and Suggestions for Authors
Pescari et al submitted a study on the roles of Trimethylamine N-oxide (TMAO) and resistin as obesity predictors, examining their associations with carotid intima-media thickness (CIMT) and metabolic parameters.
No serious plagiarism has been detected, whilst language is convenient.
Given that there is metabolism variation relative to age, it would be useful to present results in at least 2 age strata. Accordingly it would be of value to search if sexual dimorphism in risk exists.
The statistics and the presentation of the results are well performed but need to be extented by more information as noticed above.
Author Response

(The authors gave the same response as above.)

Reviewer 3 Report
Comments and Suggestions for Authors
Major Concerns & Areas for Improvement:
-
Sample Size and Study Design: The biggest weakness is the small sample size (n=60) and the observational, cross-sectional nature of the study. This severely limits the statistical power and the ability to draw robust conclusions about causality. The authors claim it's a "prospective observational study," but the description suggests it's more likely cross-sectional, as all data seems to be collected at one time point. Clarify the study design accurately. With such a small sample, especially when divided into subgroups based on BMI and TMAO levels, the statistical analyses are likely underpowered, leading to potentially unreliable results and inflated effect sizes. Recommendation: The authors should acknowledge this limitation prominently and temper their conclusions accordingly. Consider increasing the sample size substantially in future studies and adopting a longitudinal design to assess causality.
-
TMAO Threshold: The justification for the TMAO threshold of 380 mcg/L is weak. Simply stating it's "defined by the laboratory" is insufficient. What is the clinical relevance of this specific cutoff? Is it based on established literature or clinical guidelines? Provide a stronger rationale for using this particular value. Recommendation: Provide a more detailed explanation of the clinical significance of the 380 mcg/L threshold. Consider exploring the data using different cutoffs or analyzing TMAO as a continuous variable to avoid arbitrary categorization.
-
Resistin Reference Ranges: Similar to TMAO, the source and justification for the resistin reference ranges are missing. Provide the origin of these values and explain why they were used. Recommendation: Clearly state the source of the resistin reference ranges and their relevance to the study population.
-
Control Group Definition: The control group definition is problematic. Excluding individuals with any cardiometabolic risk factor, family history, smoking, or alcohol consumption creates a highly select and potentially unrepresentative group. This makes comparisons with the overweight and obese groups difficult to interpret. Recommendation: Reconsider the control group criteria. A more balanced approach might be to allow some common risk factors but control for them statistically in the analysis. Clearly define all inclusion/exclusion criteria for each group.
-
Statistical Analysis: The description of the statistical methods is very brief. What specific tests were used to compare groups and assess correlations? Were adjustments made for multiple comparisons? The logistic regression analysis needs more details. How were the variables selected for the model? What were the specific outcomes? Recommendation: Provide a more detailed description of all statistical methods used, including specific tests, adjustments for multiple comparisons, and details of the logistic regression model.
-
Missing Information: Several pieces of information are missing or unclear:
- Dietary Data: The manuscript mentions assessing dietary habits but doesn't describe how this was done (e.g., food frequency questionnaire, dietary recall). This is crucial given the influence of diet on TMAO levels.
- Physical Activity Assessment: How was "activity engagement level" quantified? What specific questionnaire or method was used? The current description is too vague.
- Sleep Duration: How was sleep duration measured? Self-reported sleep duration is often unreliable.
- Medication Use: The exclusion criteria mention excluding certain medications, but the manuscript doesn't state whether medication use was recorded for all participants. This is important as some medications can affect TMAO and resistin levels.
- CIMT Measurement Protocol: Provide more details about the CIMT measurement technique. Was it performed by a trained technician? How many measurements were taken? This is important for reproducibility.
- Ethical Approval: While ethical approval is mentioned, the specific details (e.g., IRB name, approval number) should be included.
-
Results Presentation: The results section needs to be more clearly structured and presented. The current format makes it difficult to follow the findings. Recommendation: Use tables and figures to summarize the data effectively. Report means and standard deviations (or medians and interquartile ranges if data are not normally distributed) for all relevant variables. Clearly state the statistical significance of all findings.
-
Discussion: The discussion should focus on the key findings of the study and their implications. It should also address the limitations of the study more explicitly and suggest directions for future research. The discussion should be concise and avoid excessive speculation.
-
Introduction: While the introduction provides a good overview of the background, it could be more focused on the specific research questions being addressed.
-
Keywords: The abstract mentions keywords, but they are not listed. Recommendation: Provide 3-10 relevant keywords.
Specific Examples:
- Abstract: The abstract is too long and contains too much detail. It should be more concise and focus on the main findings.
- Introduction: The discussion of BMI could be shortened. The focus should be on TMAO and resistin.
- Methods: The inclusion and exclusion criteria are poorly formatted. Use bullet points or a table for clarity.
- Results: The statement "TMAO, resistin, CIMT, fat mass, glucose, HbA1c, and lipid profiles significantly increased across BMI categories (p<0.001)" is too general. Provide specific data for each variable.
Author Response

(The authors gave the same response as above.)

Round 2
Reviewer 2 Report
Comments and Suggestions for Authors
Authors performed the analyses suggested. Yet, they comment the relevant choice and results in the Results section rather than the Discussion.
Either they merge the two sections to one (Results and Discussion), or remove the commentary to the Discussion section. In case they opt the second, they have to do so, for all their findings.
Author Response
Esteemed Reviewer,
Thank you for your valuable feedback. We have made the requested modifications by opting for the second approach. We have refined the Results section to include only the essential findings while moving the explanatory commentary to the Discussion section. This ensures a clearer distinction between the two sections and improves the overall structure of the manuscript.
We appreciate your insightful suggestions, which have helped enhance the clarity and coherence of our work.
Best regards,
Authors